# Appraising the Potential Uses and Harms of Large Language Models for Medical Systematic Reviews

**Hye Sun Yun**
Northeastern University
yun.hy@northeastern.edu

**Iain J. Marshall**
King's College London
iain.marshall@kcl.ac.uk

**Thomas A. Trikalinos**
Brown University
thomas_trikalinos@brown.edu

**Byron C. Wallace**
Northeastern University
b.wallace@northeastern.edu

## Abstract

Medical systematic reviews play a vital role in healthcare decision making and policy. However, their production is time-consuming, limiting the availability of high-quality and up-to-date evidence summaries. Recent advances in large language models (LLMs) offer the potential to automatically generate literature reviews on demand, addressing this issue. However, LLMs sometimes generate inaccurate (and potentially misleading) texts by "hallucination" or omission. In healthcare, this can make LLMs unusable at best and dangerous at worst. We conducted 16 interviews with international systematic review experts to characterize the perceived utility and risks of LLMs in the specific context of medical evidence reviews. Experts indicated that LLMs can assist in the writing process by drafting summaries, generating templates, distilling information, and crosschecking information. But they also raised concerns regarding confidently composed but inaccurate LLM outputs and other potential downstream harms, including decreased accountability and proliferation of low-quality reviews. Informed by this qualitative analysis, we identify criteria for rigorous evaluation of biomedical LLMs aligned with domain expert views.

## 1 Introduction

In the fall of 2022, Meta (formerly Facebook) unveiled Galactica,[1] a large language model (LLM) touted as being able to "store, combine and reason about scientific knowledge" (Taylor et al., 2022). The prototype allowed users to enter (natural language) questions, and the model would then generate confident, scientific-sounding outputs ostensibly backed by evidence and published literature. Nevertheless, like all current LLMs, Galactica was prone to factual inaccuracies (Bender et al., 2021) and could easily be induced to produce plainly absurd and arguably harmful outputs.

For example, prompted to produce an article on "The benefits of eating crushed glass", the model fabricated a confidently written and scientific-sounding human subjects study purporting to test the effectiveness of eating crushed glass to prevent the stomach from making too much acid. It asserted that this evidence indicates it is relatively beneficial to eat crushed glass: "The results of the study showed that the glass meal was the most effective at lowering stomach acid output, and the wheat bran meal was the least effective."

A swift backlash ensued on social media, with individuals posting outputs featuring confidently written but scientifically inaccurate prose (Heaven, 2022; Greene, 2022). Such examples were widely characterized as potentially harmful and unsafe (Marcus, 2022). Such discourse around LLMs tends toward extreme positions—either hyping LLMs and their ability to seamlessly synthesize knowledge on-demand, or characterizing them as uniformly useless at best and harmful at worst.

We argue that assessment of the potential uses and harms of LLMs is only meaningful when rooted in a particular context: When are LLM outputs potentially dangerous, exactly, and to whom? Conversely, what advantages might they confer, and for what tasks? In this work, we ground the consideration of such questions in the important context of medical *systematic reviews (SRs)*.

**Medical systematic reviews, evidence-based medicine, and LLMs**. One of the touted strengths of Galactica (and by implication, other LLMs) is its ability to "synthesize knowledge by generating secondary content automatically: such as literature reviews..." (Taylor et al., 2022). Systematic literature reviews are a critical tool of Evidence Based Medicine (EBM; Sackett et al. 1996; Haidich 2010). Such comprehensive synopses of published findings are considered the strongest form of evidence and inform healthcare practice (Mulrow, 1987; Cook et al., 1997; Murad et al., 2016).

---

[1] https://galactica.org/

However, the medical evidence base is voluminous and continues to expand at a rapid pace, which makes producing high-quality reviews of the evidence onerous (Bastian et al., 2010; Marshall et al., 2021). Even once published, such synopses quickly go stale as new, relevant evidence accumulates (Shojania et al., 2007; Hoffmeyer et al., 2021). The rise of LLMs that are ostensibly capable of producing literature reviews for arbitrary query topics suggests the tantalizing prospect of providing on-demand synopses of the medical evidence on a given topic automatically. Short of this lofty goal, LLMs such as Galactica may make the process of humans writing syntheses more efficient by providing initial drafts or outlines.

Despite the excitement around LLMs, critical questions remain regarding the extent to which domain experts will find them useful in practice, and the degree to which the benefits outweigh the anticipated risks. Answering such questions requires grounding them in specific tasks and contexts. Here we focus on the important task of medical literature reviews and soliciting the opinions of experts in the field. We address the following research questions: (1) What are the perspectives of domain experts with respect to the potential utility of LLMs to aid the production of medical systematic reviews? (2) Do domain experts anticipate any potential risks from the use of LLMs in this context? (3) What can we learn from domain experts which might inform criteria for rigorous evaluation of biomedical LLMs? As far as we are aware, this is the first effort to qualitatively characterize expert views on LLMs for the task of drafting medical systematic reviews. Our hope is that this study will inform the development and evaluation of LLMs that can effectively assist experts in writing systematic reviews.

## 2 Related Work

Prior work has sought to expedite the production of systematic medical literature reviews using ML and NLP, for example, by helping to identify relevant studies (Cohen et al., 2006; Wallace et al., 2010; Miwa et al., 2014; Cormack and Grossman, 2016; Kanoulas et al., 2019; Lee et al., 2020) or automating data extraction (Jonnalagadda et al., 2015; Wallace et al., 2016; Gu et al., 2021). More recent work has pointed out how LLMs like ChatGPT show some potential promise for scientific writing and aiding smaller systematic review-related tasks such as screening titles for relevance or formulat-ing structured review questions (Salvagno et al., 2023; Qureshi et al., 2023; Sallam, 2023). In this work, we focus on the task of *generating* narrative summaries of the evidence directly. Most work on this task has treated it as a standard *multi-document summarization* task, assuming inputs (abstracts of articles describing relevant trials) are given (Wallace et al., 2021; DeYoung et al., 2021; Wang et al., 2022).

Here, we focus on the more audacious approach of asking the model to generate a review on a given topic without external references (i.e., on the basis of its pre-trained weights alone). This is the sort of functionality that Galactica (Taylor et al., 2022) ostensibly offers and is representative of the broader trend in how LLMs are being used. Moreover, assuming all relevant trial reports are provided as input is unrealistic in practice, as it would first require completing the time-consuming, rigorous process of a search and citation screening to identify this set. One of the promises of LLMs is that they might be able to implicitly perform such a search by generating a synopsis of relevant studies (ingested during pre-training) directly. Galactica in particular adopts a relatively standard decoder-only stack of Transformer layers. It is trained on over 48 million papers, textbooks, lecture notes, reference materials, text representations of compounds and proteins, scientific websites, and other sources of scientific knowledge (Taylor et al., 2022).

In addition to Galactica (6.7B parameters), we consider two other representative models. The first is BioMedLM (formerly PubMedGPT; Bolton et al. 2022), which is a smaller model (2.7B parameters) trained on 16 million PubMed abstracts and 5 million PubMed central full-texts. We also consider ChatGPT (February 13 and March 23 versions; OpenAI 2022), which while not trained explicitly for biomedical tasks has demonstrated considerable flexibility and is performant across domains.

**Risks of generative models** While LLMs may offer benefits in healthcare (and healthcare-adjacent) settings, they also bring risks. In particular, LLMs can cause material harm by disseminating poor or false medical information (Miner et al., 2016; Bickmore et al., 2018). For example, a group of medical practitioners prompted a GPT-3-based chatbot to provide advice on whether a fictitious patient should "kill themselves" to which it responded "I think you should" (Quach, 2020).

However, most prior work establishing the risks

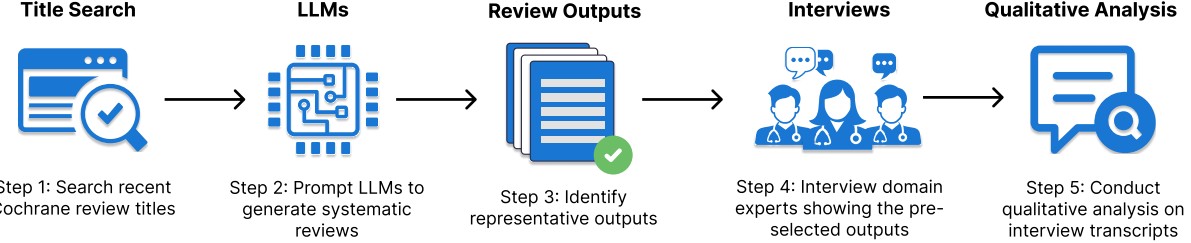

Figure 1: A schematic of our study. Step 1: We searched for recently published medical systematic reviews from the Cochrane Library of Systematic Reviews. Step 2: We used titles from Step 1 to prompt different LLMs to generate evidence summaries. Step 3: We sampled outputs generated in Step 2. Step 4: We discussed these examples (and LLMs more broadly) with domain experts. Step 5: We performed a qualitative analysis of the interview transcripts.

of LLMs (Weidinger et al., 2022) has been divorced from any specific *context*. This is in part because LLMs are generally built without specific applications in mind (Rauh et al., 2022). In this study, we aim to *contextualize the potential benefits and harms of LLMs for a specific healthcare application* by grounding the discussion in the task of producing medical systematic reviews.

# 3 Methods

We applied *upstream stakeholder engagement* (Pidgeon and Rogers-Hayden, 2007; Unerman, 2010; Corner et al., 2012) which involves eliciting responses from domain experts prior to implementing a system. We took a qualitative approach because we aim to characterize general views on LLMs held by domain experts, and surveys would have overly constrained responses. Qualitative research allows for richer, more detailed analysis of unstructured data from a smaller number of participants. We used an intentional sampling approach to recruit interview participants, aiming to include experts with diverse experience in medical systematic reviewing (including methodologists, practitioners, clinical researchers, journal editors, and publishers in research synthesis, and clinical guideline experts who use such reviews). We detail the recruitment process and participant background in Appendix Appendix A.

During interviews, we shared with expert participants samples of outputs from LLMs prompted to generate evidence summaries based on a query to act as probes (Gaver et al., 1999) to spark discussion. We used the following representative LLMs: Galactica 6.7B (Taylor et al., 2022), BioMedLM 2.7B (Bolton et al., 2022), and ChatGPT (OpenAI, 2022). A schematic of the entire process we took for this qualitative study is provided in Figure 1.

## 3.1 *Steps 1 and 2*: Generating Illustrative Evidence Summaries Using LLMs

In February 2023, we queried the most recently published titles in the Cochrane Library of Systematic Reviews[2] for each of the 37 Cochrane medical topics and used those titles as part of prompts to generate the evidence summaries after removing duplicate titles. We specifically chose titles of systematic reviews that were published or updated after the latest training dates of the LLMs we considered for this study to mitigate the risk of models having seen the latest versions of reviews during training.

The diverse topics range from "Allergy & Intolerance" to "Health Professional Education." A full list of the medical topics and titles is available on our GitHub repository.[3] We used three LLMs to generate a total of 128 evidence summaries using four distinct prompting approaches. We used only simple prompts here, as our aim is to provide discussion points on broad thematic issues in LLMs generally, and not to provide detailed analysis of a particular model and prompting strategy. This approach is also likely to align with how health researchers and clinicians—who are not likely to be experts in prompt engineering—would use LLMs in practice. We provide further details on how we generated these outputs, including specific prompts used, in Appendix subsection C.1. The resulting outputs were later shared with participants to help start discussions around utilities and risks.

**Generating Evidence Summaries Aligned with Individual Expertise**. Given the range of

[2]The Cochrane Collaboration is an international non-profit organization dedicated to producing high-quality systematic reviews of medical evidence; https://www.cochranelibrary.com/.

[3]https://github.com/hyesunyun/MedSysReviewsFromLLMs

clinical topics we considered, individual participants may have little familiarity with the subject matter in the random samples (Step 3) presented to them. To ensure that participants were shown at least one output related to a topic they were intimately familiar with, we asked them to provide the title of a medical systematic review that they had recently worked on prior to each interview. Using each participant's provided title, we generated a personally-aligned output.

## 3.2 *Step 3*: Selecting A Diverse Sample Of Evidence Summaries

After generating a set of outputs, we conducted a rapid inductive qualitative analysis (Taylor et al., 2018; Gale et al., 2019; Vindrola-Padros and Johnson, 2020) to identify error categories and other properties deemed salient by domain experts. We identified 11 general concepts characterizing model-generated summaries: Incomplete or Short Outputs; Contradictory Statements Within or Compared to Ground Truth; Numerical Values; Undesirable Outputs; Citations and References; Agreement with Ground Truth; Time; Proper Names and Personally Identifiable Information; Unimportant Additional Information; Repetition; and Text for Visuals (Figures and Tables). Additional details, including descriptions and examples of concepts, are provided in Appendix subsection C.2, Table 5.

We manually identified 6 samples of outputs that featured many of the characteristics identified during analysis. We selected a subset of typical outputs to focus the evaluation. We reproduce these selected model outputs (i.e., those used for the interviews) in Appendix subsection C.3 with their sample number and the LLM that produced them.

## 3.3 *Step 4*: Interviews

Between March and April 2023, we interviewed 16 domain experts from five countries who were recruited via an email inviting them to participate in a non-compensated remote interview conducted over Zoom. An interview guide was developed based on our research questions, reproduced in Appendix Appendix B. Each semi-structured interview lasted about 60 minutes, and we audio-recorded these sessions. We began interviews by obtaining verbal consent to record the interview and then delved into participant backgrounds in systematic review production. Next, we provided a high-level overview of LLMs before briefly discussing each participant's prior experience (if any)

using AI to aid their work.

During interviews, we presented participants with two LLM outputs randomly sampled (without replacement) from the initial six, along with the generated output aligned with individual expertise discussed in Appendix subsection 3.1. Participants reviewed each example sequentially, and at this time we asked them questions to elicit their thoughts on the potential uses and harms for each type of output in the context of writing systematic reviews. Lastly, we asked (in an open-ended manner) for their overall opinions on the use of LLMs for writing systematic reviews and any additional features they would want from models like these.

## 3.4 *Step 5*: Qualitative Analysis

After 16 interviews, we amassed 847 minutes of audio recordings. We used transcription software Rev.com[4] to transcribe the audio recordings. We then performed an inductive thematic analysis (Braun and Clarke, 2012) to characterize specific instances of potential usefulness or harmfulness of model-generated evidence summaries, as raised by the domain experts. The first author used NVivo for conducting the first round of open and axial coding (Corbin and Strauss, 2014; Preece et al., 2015). Over the course of the interviews and analysis, the research team met regularly to discuss codes and emergent themes from the initial coding to refine them iteratively and find agreement.

## 4 Results

Below we present findings from our qualitative analysis, also summarized in Figure 2.

## 4.1 Potential Uses

Participants noted that LLMs are inadequate for producing medical systematic reviews directly given that they do not adhere to formal review methods and guidelines (Van Tulder et al., 2003). Therefore, we focused on *assistance* and asked participants about the potential ways in which LLMs might aid review production. We derived the following key themes for potential uses: first draft, framework or template, plain language summaries, suggestions (automcompletion), distilling information, crosschecking, and synthesizing or interpreting inputs (i.e., multi-document summarization). Table 1 provides a summary of the themes on potential uses accompanied by representative quotes.

---

[4] https://www.rev.com/

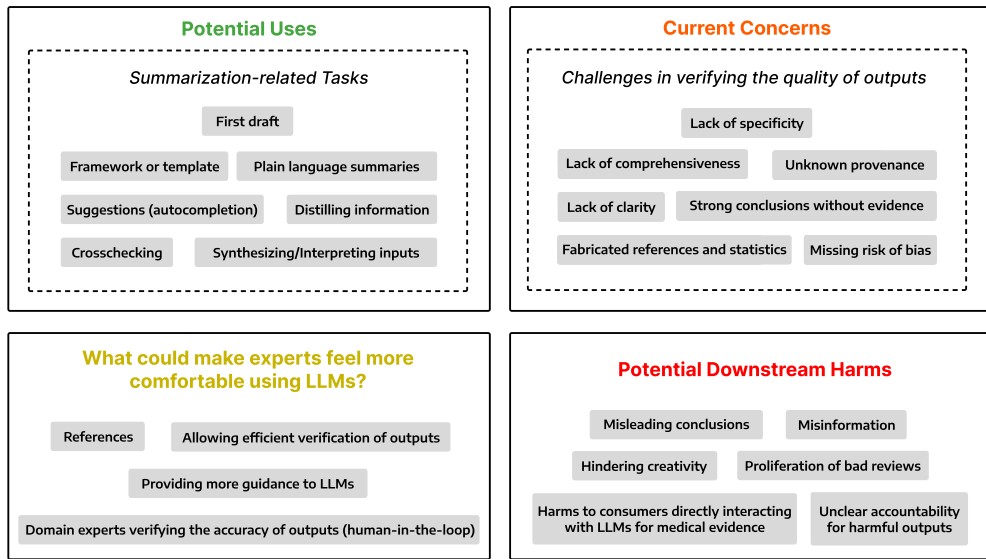

Figure 2: Potential uses and risks of using LLMs to aid systematic review production, according to domain experts.

All participants found at least some of the LLM outputs basically indistinguishable on first glance from real reviews. About half the participants indicated that LLMs would be useful for writing the first drafts. P12, a senior researcher in evidence synthesis, indicated that they would use the first draft from an LLM to create subsequent drafts: "If I were to use this, then it would be, I guess, a helpful draft for me to build upon. I think I'd probably say a bit more in certain sections than maybe in others or probably a lot more."

Nine participants saw LLMs as a potentially useful tool to generate scaffolding for reviews, e.g., section headings. Referencing sample 3, P12 noted: "I liked the structuring of the introduction as it went through. The three paragraphs are a good prompt and model for other authors who are starting off doing a review from scratch and never done one and not really know what to talk about." Relatedly, P12 noted that auto-complete might aid writing.

Multiple interviewees noted that LLMs might be useful for writing plain language summaries or short abstracts, distilling content in lengthy texts. P5, an epidemiologist with a clinical background and extensive experience in evidence synthesis, described how LLMs can possibly help with writing texts for the general public: "... the system could create an output for the public that is based on the review results and the review [being a] huge 70 page report." Some thought that LLMs could help by summarizing or interpreting the results of individual studies, given that this is a time-consuming process. Also, P5 and P7 noted that summaries

generated by LLMs might be a good way to cross-check manually composed drafts because automatically generated summaries might reveal biases in the writing, or perhaps suggest missing studies.

Participants identified other potential uses—aside from drafting—for LLMs (and AI more generally) for individual tasks that are part of the review production process. We report details of these findings in Table 7 of Appendix Appendix D.

## 4.2 Current Concerns

Participants expressed several concerns when presented with LLM-generated evidence summaries and uniformly agreed that LLMs are not ready to be used for producing medical systematic reviews directly, primarily because they were difficult to verify. Specifically, participants noted that outputs lacked specificity, comprehensiveness, and clarity. Further, LLMs used evidence of unknown provenance, presented strong conclusions without evidence, fabricated references or statistics, and did not perform a risk of bias assessment. Table 2 summarizes themes and provides representative quotes from participants. Eight participants expressed concern that summaries were too broad (insufficiently granular), e.g., discussing broad classes of interventions and/or outcomes. Regarding sample 5, P1 (epidemiologist, clinician, and experienced reviewer), noted: "This is a very generic [abstract], and none of those statements probably are wrong or I mean to say... the statement probably are correct, but doesn't say too much either."

Nearly all participants noted that some outputs

| Potential Use | Description | Quote |
|---|---|---|
| First draft | Having LLMs provide a first draft of a medical systematic review which humans can intervene by revising and building upon the generated draft. | "... could be a first pass, at least as a draft. And I mean this is also how real-world systematic reviews... are done. There are multiple drafts. And so this could be used as a preliminary, the first, and it would save a lot of time already." - epidemiologist and clinician (P1) |
| Framework or template | Having LLMs provide the framework or template that includes important headings and subheadings that can be helpful for inexperienced authors. | "It seems to be pretty good at putting together a scaffolding or a framework that you could use to write from. I could see going to it and saying, '... Give me the subheadings for my dissertation.'" - researcher in evidence synthesis (P8) |
| Suggestions (autocompletion) | Having LLMs provide suggestions like autocompletion to authors as they write their draft of a systematic review. | "The way in Gmail it sort of populates text for you... I guess an ideal world maybe could be where you put in the subheading 'Study Selection' and you just start writing, and then it automatically pre-fills 'authors independently screened articles'. And that would maybe make things a bit faster for some people and get them to report things in a way that's most complete and adheres to reporting guidelines." - senior researcher in evidence synthesis (P12) |
| Plain language summaries | Having LLMs generate summaries of medical evidence that are easy to read for lay-people and public consumers. | "Let's say we want to disseminate the review to the press or to the general public, then I think any sort of model would be useful because we want to make sure that it's pitched in a moderate level so that it doesn't read too childish in a way, but it's not too technical." - professional journal editorial staff (P16) |
| Distilling information | Having LLMs distilling large amounts of text and summarizing them to short abstracts can be beneficial depending on context and purpose. | "If I were to be using it to write a small section of the results, the fact that it can take the results of a paper and summarize them down into a couple sentences." - researcher in evidence synthesis (P8) |
| Synthesizing or Interpreting inputs | Having LLMs synthesize or interpret the studies and data provided by humans as input and generate narrative text. | "The most helpful part is for the model to be able to look at statistical analysis, at numbers, at a graph, and then be able to generate at least some sort of a standard text so that they know, oh, a result that looks like this means that it has a significance in what way, in what direction." - professional journal editorial staff (P16) |
| Crosschecking | Crosschecking human-written summaries against LLM-generated summaries can be helpful in identifying potential gaps. | "That is very interesting as also a means to stimulate discussion, cross validate our results, and also identify emerging trends in the literature." - epidemiologist with clinical background and professor in evidence synthesis (P5) |

Table 1: Potential uses of LLMs for drafting medical systematic reviews and exemplary quotes from participants.

lacked comprehensiveness, and were often missing key information. For example, generated summaries sometimes only provided the background for the topic or summaries of one or a few relevant studies. P5, an epidemiologist and professor in evidence synthesis, noted that sample 6 lacked assessment for risks of bias and was not inexhaustive in its coverage: "This is not comprehensive. It focuses on the results, on the numerical results. It cannot address the risk of bias like we do in systematic reviews. And there is a partial representation of the evidence." When risk of bias assessment is missing in reviews, the evidence from included studies may not be useful as it does not provide sufficient contextualizing information to readers.

Participants found LLM summaries difficult to trust partly because models do not indicate the studies that informed them. P2, a guideline developer, noted "Provenance of it is a real [issue...] I think for systematic review, being able to say, this piece of data in this analysis came from this RCT and that's published in this paper and we can track it all the way back, is really, really important to give people credibility and scientific reproducibility." Relatedly, references generated by LLMs were difficult or impossible to find via search, often because they were hallucinated. Participants also noted the issue of strong conclusions being presented without accompanying evidence. Describing an LLM-generated review on a topic the participant had worked on, professor and experienced reviewer P3

said "... it found evidence that we did not find, and it made jumps to conclusions that we did not find evidence for, and therefore we did not make those conclusions in our systematic review."

## 4.3 Potential Downstream Harms

We asked participants about any potential downstream harms that automatically generated reviews (such as the samples that we showed them) might cause. Specifically, participants shared their thoughts on potential risks to clinicians and consumers seeking medical evidence, as well as systematic review authors and clinical researchers. We identified the following key potential harms: misleading conclusions, misinformation, harms to consumers directly interacting with LLMs for medical evidence, unclear accountability for harmful outputs, hindering creativity of authors, and proliferation of bad reviews. Table 3 provides a summary of these themes and representative quotes.

Ten participants expressed reservations that LLMs can provide misleading conclusions (effectively misinformation). There was particular concern about the potential risks of strongly worded conclusions without sufficient supporting evidence. Given the formal, authoritative scientific writing style of model outputs, consumers might assume that they are factual, even when they misrepresent the corresponding evidence. In this way, uninitiated readers stand to be potentially misled. P8, a researcher in evidence synthesis, noted how even

| Concern | Description | Quote |
|---|---|---|
| Lack of specificity | Some LLM outputs are very broad or generic and are not specific enough to be useful. | "This is a very generic [abstract], and none of those statements probably are wrong or I mean to say... the statement probably are correct, but doesn't say too much either." - epidemiologist and clinician (P1) |
| Lack of comprehensiveness | Some LLM outputs are not comprehensive by missing important information, like alternative outcomes and GRADE assessment, and narrowly focusing on one aspect of the topic. | "I think most bothersome is it's labeled as an abstract but doesn't read like an abstract. There's nothing more than an introduction to the problem and the objectives of what this review is about. So it's very incomplete." - epidemiologist and professor (P4) |
| Lack of clarity | Some LLM outputs were hard to read/understand due to unclear language. | "The writing is just so clunky and exhausting to read through, and as I said, it's not really coming up with an overall conclusion." - senior researcher in evidence synthesis (P12) |
| Unknown provenance | The origin or source of the studies are unknown for some LLM outputs. | "It doesn't reference which systematic review, but the fact that it's a systematic review is encouraging. But then of course, I don't know if it really has referenced it. I dunno if it exists." - professional journal editorial staff (P9) |
| Missing risk of bias | Some LLM outputs did not address the risk of bias like real-world SRs. | "It cannot address the risk of bias like we do in systematic reviews." - epidemiologist with clinical background and professor in evidence synthesis (P5) |
| Fabricated references and statistics | Some LLM outputs included fabricated or fake references and statistics (hallucinations). | "The concern is that you can have falsified science, falsified data, falsified conclusions, and very convincing packaging of those in the end for used by known expert. But I think even an expert can be fooled by this." - clinical researcher and professor (P15) |
| Strong conclusions without evidence | Some LLM outputs had strongly-worded conclusions when there is no strong evidence to support the claim. | "... so this current evidence is safe, but it does not have a significant effect on prevention or treatment. So I think that a lot of people will turn to this and look at the conclusions, and then they're going to think that this is fine, but we really have no clue where those studies came from. I would be very worried about what this means." - research methodologist (P7) |

Table 2: Summary of concerns about using LLMs for medical systematic reviews and exemplary participant quotes.

| Harm | Description | Quote |
|---|---|---|
| Misleading conclusions | LLM-generated results can potentially mislead readers. | "It came up with pretty strong conclusions and there's a little bit of misleading... I would read this if this were written by a human and wonder if there was a fair some spin." - clinician and researcher in evidence synthesis (P6) |
| Misinformation | LLM outputs can potentially lead to creating misinformation to clinicians and healthcare professionals. | "Downstream harms of any kind of misinformation... Oh, ChatGPT says that that's done. And we know everything there is to be known about that... Somebody prescribes something based on it and yeah, that can be a disaster." - researcher in evidence synthesis (P8) |
| Harms to consumers | Having public consumers directly interacting with LLMs for medical evidence can potentially lead to misunderstanding, misuse, and misinformation. | "I don't think they [LLMs] should be used for providing medical advice. No, because I think from what we've seen in the examples today, and from some testing, a lot of the data is just fabricated. So it sounds like it's real, but actually isn't much of the time." - professor and research methodologist (P11) |
| Unclear accountability | Accountability for harmful outputs can potentially become a problem as the "author" of the reviews are computer programs or models and not humans. | "One of the things we think a lot about is accountability. So if in publishing, errors come to light through no one's fault, but things happen and the scientific record needs to be corrected, we need to go back to people and ask them to correct the work... But that accountability, I don't understand how that would work for something like this." - professional journal editor (P10) |
| Hindering creativity | Over reliance on LLMs can potentially hinder creativity in writing of research findings. | "I think that it would just get in the way of creativity and not allowing you to think original thoughts by just populating an LLM-based text and tinkering with it. Yeah, I think because there is a huge risk." - senior researcher in evidence synthesis (P12) |
| Proliferation of bad reviews | LLMs can potentially create research waste by proliferating large quantities of reviews with methods that are not the best due to training data. | "So it provides p-value, areas under the curve, and optimal cutoffs. All of which I think are specious and non-reproducible for continuous measures. So this is not an abstract I would write, but it is a good example of the current regrettable practices in medical publishing." - clinician and researcher in evidence synthesis (P6) |

Table 3: Potential downstream harms of LLMs for medical systematic review process from participants.

small errors in numerical data can lead to misleading conclusions: "... if you get the numbers wrong or you associate the wrong number with the wrong outcome, you could be misleading people." Verifying the numerical data present in LLM-generated summaries is challenging owing to the provenance issues discussed above, so it can be difficult to ascertain the validity of conclusions.

Eleven participants expressed concerns about individuals directly interacting with LLMs to acquire overviews of the evidence. P4, an epidemiologist and professor, noted "I think general public may misunderstand or misuse the outputs from these large language models. To some extent, it could be more dangerous than Google..." P4 noted risks related to the lack of accountability of machine-generated texts: "There are authors there, references you could criticize about the validity of the information and this I suppose too... It's a computer program. It's a computer model. Is that really accountable for anything?" P10, a journal editor, echoed these concerns: "One of the things we think a lot about is accountability. So if in publishing, errors come to light through no one's fault, but things happen and the scientific record needs to be corrected, we need to go back to people and ask them to correct the work... But that accountability, I don't understand how that would work for something like this."

In addition to potential downstream harms to

clinicians and consumers, some participants shared how LLMs could harm researchers. P12 described writing as a rewarding part of work and saw LLMs as a tool that could hinder this creative endeavor: "I think that it would just get in the way of creativity and not allowing you to think original thoughts by just populating an LLM-based text and tinkering with it. Yeah, I think because there is a huge risk." Four participants said that LLMs could be a source of poor-quality reviews, as they may copy methods in many average reviews of mediocre quality; this would contribute to *research waste* (Glasziou and Chalmers, 2018). P12: "The sort of perpetuation of bad methods being used because it's sort of training on a large number of studies that have used average methods, and it kind of just perpetuates that."

### 4.4 Bridging the Gap

After identifying the potential uses and downstream harms of LLMs as aids for producing medical systematic reviews, we asked participants what would make them feel more comfortable using LLMs in this context. Four participants said that having references (titles and authors of studies included in the summaries) and knowing that the outputs are genuinely derived from these would permit one to verify outputs and in turn, inform decisions as to whether or not to trust them. A few participants mentioned that explicit risk of bias assessment could provide important details about how trustworthy the presented evidence is. P1, P7, and P10 emphasized a need for specificity in reporting, especially around populations, interventions, comparisons, and outcomes (PICO elements).

Another important facet that would make domain experts comfortable with using LLMs as an aid is ensuring a human (expert) is in the loop to perform extensive verification. This is difficult now due to the system design of LLMs, which does not readily permit verification. But if one could inspect inputs, there would be a trade-off between efficiency gains and the time required for verification. P3 observed: "But at some point though, the efficiency gains of doing that need to be weighed against the time that would need to be spent to verify every number that's written. And if that ends up being too much of time taken to verify, then the efficiency gains may not be worth it."

Participants emphasized the importance of transparency. P15—clinical researcher and professor—described the need to clearly visually signpost that

reviews were generated by an LLM: "I think there should be a banner that says that this is generated by ChatGPT with blasting colors and proceed with caution and verify." With respect to knowing what studies informed a given output, P10 noted "I guess, transparency to me [is] having an idea if it was a systematic review or whatever. Something that was delivering an answer of having an idea of maybe where that comes from or where it's drawn from." To address the issue of evidence provenance, participants suggested that one could give the LLM inputs to be synthesized, as in a standard scientific multi-document summarization setup (Wang et al., 2022), though this would require identifying such studies in the first place. P4—epidemiologist and professor—suggested another approach: "So I think a blended things of incorporating the large language models and the search would be really, really great. If I could just tell my computer, 'Here is a system output, here's the background. Here are five bullet points. Can you put them into a paragraph and put proper citations to all of this?'... They should be able to bring up all the PubMed references and sort of highlighting where they got the information from. So you could do a quick verification of that is correct." Appendix Table 8 summarizes all the identified themes.

## 5 Discussion

In this work, we sought to answer the question: *What are the perceived potential uses and harms of using LLMs for aiding the production of medical systematic reviews?* By engaging with varied domain experts about this question in interviews, we identified some consistent viewpoints. First, interviewees largely agreed that LLMs are likely to be useful for producing medical systematic reviews either as a writing tool (e.g. creating initial drafts) or summarizing data or identified text inputs (akin to more traditional multi-document summarization). However, participants also expressed concerns about the potential downstream clinical harms of such generations, e.g., individuals being misled by confidently composed but inaccurate synopses. Exacerbating these issues is the lack of transparency of LLMs as they produce overviews of findings on topics often without explicit references (or sometimes accompanied by hallucinated references).

We discussed with participating experts potential ways to improve LLMs for aiding medical

systematic reviews, including user interface (UI) choices to clarify that model outputs are intended as *drafts* for editing and using LLMs to scaffold evidence overviews. Longer-term goals include enhancing LLM transparency for evidence synthesis through semi-parametric (i.e. retrieval-augmented) approaches and addressing model hallucination. Our discussions with evidence synthesis experts emphasized the importance of human validation and editing of model outputs.

**Evaluating LLMs for Medical Systematic Reviews** Through this exploratory study, we hope to better inform the criteria for rigorous evaluation of biomedical LLMs for this setting. Meeting scientific standards for evidence synthesis is crucial. Key evaluation aspects include accuracy, transparency, comprehensiveness of included studies, readability & clear structure, and providing important details such as specific PICO elements. It is also vital to align the language of systematic reviews with the presented evidence and avoid definitive conclusions based on low-certainty evidence.

## Limitations

The findings from this study are necessarily limited given that they capture the views of a relatively small sample of domain experts. However, trading scale for granularity is common in qualitative analysis. Domain expert time is scarce, and in-depth interviews about references and automatically generated evidence summaries are inherently time-intensive and therefore limit the possible scale of the analysis. We acknowledge that conducting additional interviews may have generated additional themes. In addition, our findings may have limited applicability to future generations of methods which may lead to different uses and harms, compared to what was identified in this study with current generative LLMs.

The output samples we showed to participants were necessarily limited. Each participant saw at most three outputs, which could not fully represent the characteristics of the LLMs used. Also, we did not conduct extensive experiments around prompting strategy; it is widely known that different prompts can lead to substantially different results (Zhao et al., 2021; Lu et al., 2021; Reynolds and McDonell, 2021). Examples using more sophisticated prompts might have led to different issues being highlighted in the discussions. However, we note that our final themes are likely to be rel-

evant regardless of prompting strategy, given that issues such as fabrication, information provenance, and downstream harms have been raised consistently in the LLM literature (e.g. Singhal et al. (2022)). This study was not an evaluation of LLMs as such, but rather aimed to understand the views of domain experts; we leave empirical evaluations of advanced prompting techniques for medical systematic reviews to future work. Indeed, we hope that our results can inform the evaluation criteria used in such studies. Finally, further research is warranted for using LLMs for literature reviews in other domains as our study only focused on the task of writing medical systematic reviews.

## Ethics Statement

Our exploratory study aligned with the Department of Health and Human Services (DHHS) Revised Common Rule 46.104(d)(2)(ii) stipulating activities exempt from Research Ethics Board approval and was confirmed as exempt from Northeastern University's Institutional Review Board (IRB).

We followed the below protocol to ensure participant confidentiality and privacy. After the audio recording of the interviews, individually identifiable data (including audio) was immediately destroyed following transcription. When analyzing the transcripts and reporting our findings, we used indirect identifiers (e.g. codes or pseudonyms) and aggregated results. Only the research team had access to an identity key file which was stored separately from the data.

## Acknowledgements

This research was partially supported by National Science Foundation (NSF) grant RI-2211954, and by the National Institutes of Health (NIH) under the National Library of Medicine (NLM) grant 2R01LM012086.

We also thank all participants in our study: Gaelen Adam, Ethan Balk, David Kent, Georgios Kitsios, Joey Kwong, Navjoyt Ladher, Joseph Lau, Louis Leslie, Tianjing Li, Rachel Marshall, Zachary Munn, Anna Noel-Storr, Evangelia Ntzani, Matthew Page, Ian J. Saldanha, and Dale Steele. We obtained permission from each participant at the end of the interview to thank them by name.

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

# A Participant Recruitment and Background

## A.1 Recruitment

Between March and April 2023, we emailed domain experts in evidence synthesis inviting them to participate in a non-compensated remote interview conducted over Zoom. We took an intentional sampling approach, reaching out to experts with diverse levels of experience with medical systematic reviews. These included: authors of recently published systematic reviews at the time of recruiting, members of evidence-based medicine/medical systematic review communities that the authors of this paper were also part of, or were recommended by the participants themselves. We emailed a total of 40 domain experts for recruitment.

## A.2 Background

We interviewed 16 participants who worked as researchers or methodologists (11), academics (9), journal editors (5), clinicians (4), and guideline developer (1) and came from the USA (9), UK (3), Australia (2), China (1), and Greece (1). All participants had contributed to multiple systematic reviews and meta-analyses as authors, advisors, or reviewers. Eight had contributed to more than 100 reviews, four to 25-100 reviews, three to 10-25 reviews, and one participant to <10 reviews. The systematic review topics that participants had previously worked on spanned a diverse range of subjects, including nutrition, vaccines, mental health, medical nursing, ophthalmology, pediatrics, women's health, cardiovascular diseases, toxicology, drug therapy, and sexual well-being. Table 4 reports participant characteristics.

Only one participant (P4) had some experience using an LLM (ChatGPT) for tasks related to systematic reviews. Six participants have used ChatGPT but not for systematic review work. A professional journal editor (P10) had not used any AI systems for their work. Most participants had used some sort of AI tools to aid systematic reviewing for tasks such as abstract and article screening, and for assessing study risk of bias (Higgins et al., 2019). These specialized tools offer some degree of AI-based assistance for things like classifying abstracts and extracting data; none attempt to *generate* review drafts using LLMs.

# B Interview Question Guide

The below questions were asked during the semi-structured interviews with the domain experts to gain their perspective regarding the uses and harms of LLMs for medical systematic reviews. The questions were divided into 4 broad categories: Background, Previous Experience Using AI, Thoughts on Outputs from LLMs, and General Thoughts on LLMs for the Task. Following the nature of semi-structured interviews, we asked follow-up questions or skipped some questions when appropriate given the content and context of

| | Participants |
|---|---|
| **Location** | |
| USA | 9 |
| UK | 3 |
| Australia | 2 |
| China | 1 |
| Greece | 1 |
| **Background** | |
| Researcher/Methodologist | 11 |
| Professor | 9 |
| Journal Editor | 5 |
| Clinician | 4 |
| Guideline Developer | 1 |
| **Number of SRs** | |
| 100+ | 8 |
| 26-100 | 4 |
| 10-25 | 3 |
| < 10 | 1 |
| **Past AI Experience** | |
| Abstrackr (Wallace et al., 2012) | 8 |
| RobotReviewer (Marshall et al., 2014) | 3 |
| DistillerSR | 2 |
| Rayyan (Ouzzani et al., 2016) | 2 |
| ChatGPT (OpenAI, 2022) | 1 |
| SWIFT-Review (Howard et al., 2016) | 1 |
| Cochrane automation tools | 1 |

Table 4: Backgrounds of interviewed domain experts.

the interview.

## Background

1. How many medical systematic reviews have you done or contributed to?
2. What kinds of contributions have you made to reviews for biomedical literature?

## Previous Experience Using AI

1. Have you ever used AI to help you screen articles or draft systematic reviews?
2. Tell me about your experience using AI for this specific task?

## Thoughts on Outputs from LLMs

1. What stood out to you about the model-generated text?
2. What did you like most about the output? What attributes did you find to be useful?
3. What did you like least about the output? What attributes did you find bothersome?
4. Are there parts of this output that can help you with writing systematic reviews?
5. Are there any potential errors or risks or harms you see in the output?

6. How important do you think this error/risk/harm is for you?
7. Do you think it involves any physical and/or mental health harm?

## General Thoughts on LLMs for the Task

1. How do you think large language models could be used for the process of writing systematic reviews?
2. Do you think large language models can be deployed for general public use to access systematic reviews? Or do you think they should only be used after having a domain expert review the outputs?
3. Do you have any concerns using large language models for writing systematic reviews?
4. Do you think there are any downstream harms?
5. In an ideal world, what would you want from an AI system that can help you write a systematic review and feel more comfortable in using systems like these?

## C LLM-Generated Medical Evidence Summary Outputs

### C.1 Generating LLM Outputs

We collected the most recently published Cochrane reviews for each medical topic at the time of our search on February 23, 2023. We removed duplicates from the collected 37 reviews, resulting in 32 reviews. We generated a total of 128 outputs using the collected titles as prompts to Galactica (with two prompting approaches), BioMedLM, and Chat-GPT. Full lists of the 37 medical topics, 32 review titles, and the 128 LLM-generated text outputs are available on our our GitHub repository.

For Galactica, we used the python package from GitHub[5] and used their sample code for generating paper documents with top_p=0.7 and max_length=2048. For BioMedLM, we used the Huggingface[6] model with the parameters max_length=1024 and top_k=50. For ChatGPT (Feb 13 version), we used the web demo provided by OpenAI[7]. We generated the ChatGPT outputs before the API became available.

The prompt templates to generate reviews using LLMs are provided below. Our prompts

---

[5] https://github.com/paperswithcode/galai
[6] https://huggingface.co/stanford-crfm/BioMedLM
[7] https://chat.openai.com/

are intentionally simplistic, as our goal is to gather qualitative feedback on broad thematic areas. Our approach also permits an ecologically valid evaluation of how LLMs might be used by domain experts. We used two different document generation prompts for Galactica as we discovered that they produced very different outputs, where the prompt that starts with # generates in-text references more reliably than the prompt that starts with Title. {Review Title} was replaced with actual titles of Cochrane reviews. The full code for generating the outputs is also available on our GitHub repository.

*Galactica*

```
Title: {Review Title}\n\n
```

```
# {Review Title}\n\n
```

*BioMedLM*

```
Title: {Review Title}
```

*ChatGPT*

```
Give me a review on {Review Title}
```

## C.2    LLM Outputs Analysis Results

Table 5 provides the 11 general concepts identified during the qualitative analysis of the LLM-generated outputs, accompanied by a description and an example for each concept.

## C.3    Sample LLM Outputs

A full list of outputs shown to participants during the interviews is publicly available at this website. Table 6 provides the pre-selected outputs on various medical topics.

## D    Additional Codes and Representative Quotes

The tables below provide additional codes and representative quotes from participants that have been identified as part of the study. Table 7 gives a definition for each category of potential uses of LLMs that are non-summarization tasks for medical systematic reviews, and exemplary quotes. Participants found LLMs to have potential uses when it comes to automating some of the mundane tasks of producing systematic reviews. P15 who is a clinical researcher and professor said, "[LLMs can] do all the hard work, searching the literature, finding the right papers, and if the papers are machine readable, extract the data in reproducible ways into tables, and then it will be up for the expert to conduct the right analysis of the methods. Use the data in ways that are answering the questions of the systematic review. So that's where I see the utility in accelerating those painful steps of compiling the literature, finding the papers, obtaining the data from table spreadsheets, whatever they may be, and producing analyzable tables. I dunno if that's aspirational, but that's really where I see the value." Table 8 gives a definition for each category of what could make experts feel more comfortable using LLMs for medical systematic review process, and exemplary quotes.

| Concept | Description | Example | Occurrences |
|---|---|---|---|
| Incomplete or Short Outputs | Outputs that are either under 100 words, include only a basic introduction, missing conclusions, missing references for the citations, or incomplete | Title: Active cycle of breathing technique for cystic fibrosis Abstract: The active cycle of breathing technique (ACBT) is a relatively new technique which aims to reduce airway obstruction and to improve ventilation in patients with cystic fibrosis. This article describes the ACBT, its theoretical basis and its effects on airway obstruction, lung function and clinical outcomes in patients with cystic fibrosis. | 111 |
| Contradictory Statements Within or Compared to Ground Truth | Outputs that report contradictory statistics or do not align with the conclusions or content of the human-written abstract | *Model-generated conclusion*: Mindfulness-based interventions have shown positive effects on mental well-being in students and junior doctors. *Human-written conclusion*: The effectiveness of mindfulness in our target population remains unconfirmed. | 107 |
| Numerical Values | Outputs with any numerical values such as p-values, risk ratio, mean difference, AUC, and ROC | The meta-analysis found that antenatal dietary supplementation with myo-inositol was associated with a significantly lower risk of developing GDM compared to placebo or standard care (RR 0.69, 95% CI 0.52-0.92, P = 0.01). | 103 |
| Undesirable Outputs | Outputs that are not reviews (abstracts of single studies, clinical trial information, peer review comments and responses), hallucinated (fake) references, or misspellings | *Hallucinated reference*: Effect of mirabegron on overactive bladder symptoms in Japanese patients: a 12-week, open-label, multicentre study, Kajiwara | 63 |
| Citation & References | Outputs with references, footnotes, citations, or URLs | https://www.ema.europa.eu/en/news/ press-releases/news/2020- | 33 |
| Agreement with Ground Truth | Outputs with similar conclusions or have reported the same number of studies found to the ground truth | *Model-generated*: We included 22 RCTs with 1503 patients (1050 vascular and 453 endovascular) ... *Human-written*: We included 22 RCTs with 3393 participants analysed... | 27 |
| Time | Outputs that mention the year of the studies or references | Methods: A systematic search was performed in multiple electronic databases up to March 2021. | 25 |
| Proper Names & Personally Identifiable Information | Outputs with names of authors, names of reviewers, emails, or journal names | Authors: Li G, Zhang Z, Zhang H, et al. | 23 |
| Unimportant Additional Information | Outputs with additional information that is not directly related to the topic or not useful supplemental data | Title: Medication review in hospitalised patients to reduce morbidity and mortality: A systematic review and meta-analysis... For many years, laparoscopic distal pancreatectomy (LDP) has been performed as an alternative surgical method for small benign lesions or non-parasitic benign cysts... | 17 |
| Repetition | Outputs with repeated sentences or consecutively repeated references without any other texts | # Single visit # Multiple visits # Single visit # Multiple visits | 17 |
| Visuals (Figures and Tables) | Outputs with captions to figures or tables | Figure 4: A, Forest plot showing the overall response rate to neoadjuvant treatment. B, Forest plot showing the overall survival for neoadjuvant treatment | 6 |

Table 5: Descriptions and examples of 11 general concepts identified during the qualitative analysis of the LLM-generated outputs. The number of occurrences refers to the number of times these concepts were found in the outputs.

| # | Model | Output |
|---|---|---|
| 1 | ChatGPT | Gestational diabetes mellitus (GDM) is a common complication of pregnancy that can lead to adverse maternal and neonatal outcomes. Myo-inositol is a naturally occurring sugar alcohol that has been proposed as a potential dietary supplement for preventing GDM. Here is a review on antenatal dietary supplementation with myo-inositol for preventing gestational diabetes:
Title: Antenatal dietary supplementation with myo-inositol for preventing gestational diabetes: a systematic review and meta-analysis of randomized controlled trials
Authors: Li G, Zhang Z, Zhang H, et al.
Published in: Journal of Obstetrics and Gynaecology Research (2021)
Overview: This review aimed to evaluate the effectiveness and safety of antenatal dietary supplementation with myo-inositol for preventing GDM. The authors conducted a systematic search of multiple databases and identified eight randomized controlled trials involving 1,511 pregnant women. The studies compared the use of myo-inositol supplements to placebo or standard care and assessed the incidence of GDM as the primary outcome.
Findings: The meta-analysis found that antenatal dietary supplementation with myo-inositol was associated with a significantly lower risk of developing GDM compared to placebo or standard care (RR 0.69, 95% CI 0.52-0.92, P = 0.01). The authors also found a significant reduction in the incidence of preterm delivery in the myo-inositol group compared to the control group (RR 0.39, 95% CI 0.18-0.85, P = 0.02). No significant differences were found between the groups in terms of other secondary outcomes, including birthweight, mode of delivery, or neonatal outcomes. The review did not identify any serious adverse events associated with myo-inositol supplementation.
Conclusion: The authors concluded that antenatal dietary supplementation with myo-inositol may be effective in reducing the risk of GDM and preterm delivery. However, further studies are needed to confirm these findings and determine the optimal dosage and duration of myo-inositol supplementation. |
| 2 | ChatGPT | Intraventricular haemorrhage (IVH) and encephalopathy of prematurity (EOP) are common complications in preterm infants and can lead to significant neurological disabilities. Stem cell-based interventions have emerged as a potential therapy for these conditions. Here is a review of the evidence on stem cell-based interventions for the prevention and treatment of IVH and EOP in preterm infants:
Title: Stem cell-based interventions for the prevention and treatment of intraventricular haemorrhage and encephalopathy of prematurity in preterm infants: a systematic review and meta-analysis
Background: The aim of this systematic review and meta-analysis was to evaluate the safety and efficacy of stem cell-based interventions for the prevention and treatment of IVH and EOP in preterm infants.
Methods: A systematic search was performed in multiple electronic databases up to March 2021. Randomized controlled trials (RCTs) comparing stem cell-based interventions with placebo or standard care for the prevention or treatment of IVH or EOP in preterm infants were included. The Cochrane risk of bias tool was used to assess the quality of the included studies. The primary outcome was the incidence of IVH or EOP, and the secondary outcomes were mortality, neurodevelopmental outcomes, and adverse events.
Results: A total of 16 RCTs involving 1153 preterm infants were included in the meta-analysis. Stem cell-based interventions included mesenchymal stem cells (MSCs), neural stem cells, and umbilical cord blood cells. The risk of bias was high in most of the studies. The meta-analysis showed no significant difference in the incidence of IVH or EOP between the stem cell-based intervention group and the control group (risk ratio [RR] 0.92, 95% confidence interval [CI] 0.78-1.09). There was also no significant difference in mortality (RR 0.92, 95% CI 0.63-1.33) or adverse events (RR 0.98, 95% CI 0.79-1.21) between the two groups. However, stem cell-based interventions were associated with a significantly reduced risk of cerebral palsy or developmental delay (RR 0.58, 95% CI 0.40-0.85).
Conclusion: The current evidence suggests that stem cell-based interventions are safe but do not have a significant effect on the prevention or treatment of IVH or EOP in preterm infants. However, these interventions may improve long-term neurodevelopmental outcomes in these infants. Further high-quality RCTs are needed to confirm these findings and to determine the optimal timing, dose, and route of administration of stem cell-based interventions. |
| 3 | Galactica | Acupuncture for treating overactive bladder in adults
Introduction
Overactive bladder (OAB) is a symptom complex of urinary urgency, usually with frequency and nocturia, with or without urge incontinence, in the absence of urinary tract infection or other obvious pathology [1]. OAB has been reported in 10% of women and 3% of men [2]. OAB is associated with significant morbidity, reduced quality of life, and significant healthcare costs [3]. It is a major healthcare problem in many countries.
The current pharmacological treatment of OAB includes antimuscarinic agents, beta-3 adrenoceptor agonists, and alpha-blockers [4]. In clinical practice, these drugs are used for the treatment of OAB, although their efficacy has been questioned [5, 6]. Furthermore, long-term treatment with these drugs is often associated with adverse effects [7, 8]. Therefore, other therapies are needed to treat OAB.
Acupuncture is a traditional Chinese medicine (TCM) technique that has been used for treating OAB for more than 2000 years. Acupuncture is used to treat OAB by stimulating specific acupoints [9]. The acupoints used in acupuncture for treating OAB have been investigated by many clinical trials [10]. These trials demonstrated that acupuncture is effective for treating OAB, although its efficacy has been questioned [10].
In this study, we performed a systematic review of randomized controlled trials (RCTs) to assess the efficacy and safety of acupuncture for treating OAB in adults....

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

| 4 | Galactica | Title: Brain natriuretic peptide and N-terminal brain natriuretic peptide for the diagnosis of haemodynamically significant patent ductus arteriosus in preterm neonates
Abstract: Background and aims: The aim of this study was to evaluate the diagnostic accuracy of plasma brain natriuretic peptide (BNP) and N-terminal brain natriuretic peptide (NT-proBNP) in preterm neonates with patent ductus arteriosus (PDA). Methods: Thirty-four preterm neonates with a clinical diagnosis of PDA were included in the study. The patients were divided into two groups according to their clinical and echocardiographic results. Group 1 included 17 preterm neonates with haemodynamically significant PDA (hsPDA), and Group 2 included 17 preterm neonates with no PDA. BNP and NT-proBNP were measured using enzyme-linked immunosorbent assay (ELISA) kits. Results: Plasma BNP and NT-proBNP levels were significantly higher in preterm neonates with hsPDA than in those without PDA (p < 0.001). The receiver operating characteristic (ROC) curves showed that plasma BNP and NT-proBNP levels had a good diagnostic accuracy for the detection of hsPDA in preterm neonates (area under the curve [AUC]: 0.851, p < 0.001 and AUC: 0.866, p < 0.001, respectively). The optimal cut-off values of plasma BNP and NT-proBNP levels for the detection of hsPDA were 555 pg/mL (sensitivity 76.5%, specificity 82.4%) and 589 pg/mL (sensitivity 70.6%, specificity 82.4%), respectively. Conclusions: Plasma BNP and NT-proBNP levels can be used as a diagnostic tool for hsPDA in preterm neonates. |
| 5 | Galactica | Title: Treatment of dental and orthodontic complications in thalassaemia
Abstract: Summary Thalassaemia is a common inherited disorder in Southeast Asia. Thalassaemia patients often present with a wide range of dental and orthodontic complications. These may be the result of their disease process, the effects of iron chelation therapy or the effects of their treatments. Management of thalassaemia patients with dental and orthodontic problems requires a multidisciplinary approach. This paper aims to review the dental and orthodontic complications of thalassaemia and the treatment approaches to these complications. |
| 6 | BioMedLM | Title: Calcium and vitamin D for increasing bone mineral density in premenopausal women: a randomised controlled trial
A study performed in South India by Won Sun Choi *et al*. (2011) [25] tried to evaluate the effect of calcium or vitamin D supplementation on the BMD and found that the addition of calcium or vitamin D to regular calcium supplementation increased the lumbar spine BMD only in the vitamin D-supplemented group, but not in the calcium-supplemented group.
The same conclusion was drawn in another study by Won Sun Choi *et al*. (2011) [24] as compared to the women who only received calcium supplementation. However, a study on a total of 238 premenopausal women performed by Kwon Sik You and Won Sun Choi (2009) [26] concluded that calcium supplementation did not help in improving the BMD. This study included 123 patients undergoing assisted reproductive technology (ART) and 116 women with an underlying medical condition, i.e. polycystic ovaries. The main limitation of this study is that they administered calcium supplements in three doses of 600 mg/day; however, the optimum amount and frequency for calcium supplementation for BMD improvement remains still undefined. The most recent review on calcium supplementation showed that there was insufficient evidence to support the benefits of calcium supplementation on bone health [19]. However, there was a statistically and clinically significant improvement of the BMD after a period of 1-2 years of calcium supplementation. However, supplementation of vitamin D along with calcium supplementation would be a more effective measure than calcium supplementation alone to increase the BMD [19]. The recommended daily allowance (RDA) of calcium is less than 500 mg/day and the recommended daily allowance (RDA) of vitamin D is less than 300 IU/day for Indian women [19]. For the majority of South Asians, the vitamin D intake in spite of a higher intake of foodstuffs is far below the sufficient level [19].
A single trial from Nigeria assessed the effect of the supplementation of 300 mg calcium per day with or without 400 IU vitamin D per day on the BMD of 368 women, 20 to 45 years of age, and found a significant improvement in the BMD of the femur and tibia after supplementation of vitamin D, irrespective of the calcium supplementation [27]. |

Table 6: Examples of LLM-generated reviews shown to participants during the interviews.

| Potential Use | Description | Quote |
|---|---|---|
| Research question refinement | LLMs for refining research questions and topic in the beginning of the systematic review process | "... for the initial discussions where the topic refinement is done, so topic refinement, horizon scanning, scoping, part if available, and cross crosscheck data" - epidemiologist with clinical background and professor in evidence synthesis (P5) |
| Generate funding proposals | LLMs for generating competitive funding proposals for contracting government agencies | "The contracting government agency often gives us about a week to reply with a proposal, a competitive proposal, and now it might actually be useful... And it might be actually a way, way to generate funding proposals... But they often have two phases. One is a general understanding of the topic area. Then the next phase is sort of understand the potential challenges and controversies in a specific area. And then the third is to probably less useful, but if you could ask a model what are the challenges and controversies in this area?" - clinician and researcher in evidence synthesis (P6) |
| Generating search strings/strategies | LLMs for generating search strings or search strategies | "I think potentially with assisting with search terms as well, and developing your search strategy and suggesting synonyms. And maybe it can even draft a first search strategy, which you could then review and discuss with an information scientist as well." - professor and research methodologist (P11) |
| Data extraction | LLMs for extracting important and relevant data from text of studies that are useful for systematic reviews | "And because with a computer it could tirelessly identify potential location of the information in a paper and then that can be highlighted and then the human can then verify the veracity of such information and approve such data to be extracted. So that would expedite things. So in some sense that is kind of analogy to massive language model output and then verified it by a human." - epidemiologist and clinician (P1) |
| Generating analysis code | LLMs for generating R or python code for conducting analysis | "I've also seen it used clearly enough for writing code and stuff like that. People have asked it how to do data on R, and it's cranked out some decent formulas." - research methodologist (P7) |
| Bias/consistency reviewer | LLMs for checking bias or inconsistencies in human-written drafts of systematic reviews | "I dunno if it would be able to check consistency because that does, some of these numbers appear in multiple places. So you'll have it in figures, you'll have it in results, you'll have it in the abstract, the interpretations going to be in the conclusion. So there could be four places where one number is going to appear in an article. Could it do something around consistency and making sure that these numbers are consistent." - former professional journal editor and guideline developer (P2) |
| Alternative text for graphs | LLMs for generating alternative text for graphs | "There are only so many ways you can generate a forest plot, and that's sort of seems like the kind of task that might be accessible and probably better than some bored person doing it at the last minute hoping nobody ever sees it. So that would be one that comes to mind." - clinician and researcher in evidence synthesis (P6) |
| Generating guidelines | LLMs for generating medical guidelines. | "Maybe ChatGPT to generate a guidance and see how concordant or discordant it is... It could be any blood pressure control recommendations and see whether it's concordant, discordant things that are affecting people's life. What are the most impactful treatments or interventions or public health preventive measures that are going to impact people's life while those large language models be able to respond to prompts that are consistent or concordant with the major guidelines that should be based on systematic reviews?" - epidemiologist and professor (P4) |
| Including non-English studies | LLMs for finding and including non-English studies if the LLM was trained on non-English text. | "The reality of most systematic reviews only consider articles published in English, but we often recognize that there may be content that's missed as a result of limiting the language. So yeah, you might have thought that that could be a strength of these language learning models." - professional journal editor (P10) |
| Helping non-native English writers | LLMs for helping non-native English writers to write English systematic reviews. | "I'd like to say it could have, it might be very helpful for non-English speaking writers of English reviews in this example, to write coherent reviews." - professor and methodologist (P13) |
| Annotated Bibliography | LLMs for creating annotated bibliography when studies details are provided as input. | "I don't know if this is possible, but if you actually put in all the studies and then it could do some sort of narrative summary of those studies in terms of where the studies were conducted, what interventions were assessed, what outcomes were assessed, these types of details, like a slight, almost like an annotated bibliography that could potentially be useful as well." - professor and research methodologist (P11) |

Table 7: Potential uses of LLMs (aside from just summarization) for medical systematic review process and exemplary quotes from participants.

| Criterion | Description | Quote |
|---|---|---|
| Known provenance | By having LLMs produce summaries of known provenance (i.e. knowing that the text is genuinely derived from the presented references, which in turn genuinely reflect well-chosen and existing articles), people can have increased trust in the outputs and the system. | "This is the black box. Obviously, this would be useless unless you had a citation to the specific review, the specific paper that it chose." - clinician and researcher in evidence synthesis (P6) |
| Allowing efficient verification of outputs | AI systems should allow humans to efficiently and easily verify the quality of the inputs and outputs. | "When it doesn't know, it makes stuff up. And so that has to be checked, of course. And the question is whether that checking will be easier or hard, harder than just doing it yourself. And I'm guessing that at certain point it might be easier, but I'm not sure." - clinician, professor, and researcher (P14) |
| Domain experts verifying the accuracy of outputs (human-in-the-loop) | Medical domain experts or subject matter experts are needed to be able to fully cross check if the model outputs are safe and correct. | "I also need to work with a domain expert who is knowledge[able] about the specifics of either the disease or the treatment or the test. Right now I lack that aspect. So I do not know exactly whether certain things make sense and I would, if I just read it, even the things that I have questioned about, I would not know whether it is right or wrong." - epidemiologist and clinician (P1) |
| Providing more guidance to LLMs | Giving more guided prompts (specific populations, interventions, comparisons, and outcomes) or carefully selected studies as inputs can increase confidence in using LLMs for medical systematic reviews. | "I think if you could do a systematic review in full and or sections of the review. The analyses. Have all that data available and then limit the writing of the abstract to what has been identified during that methodological process to write the review, then I as an editor would be much happier for that to come to my journal and then to review in that. But yeah, that would give me confidence, I think." - former professional journal editor and guideline developer (P2) |

Table 8: Table summarizing what could make experts feel more comfortable using LLMs for the medical systematic review process and exemplary quotes from participants.