# OpenReview forum: "Appraising the Potential Uses and Harms of LLMs for Medical Systematic Reviews"
_EMNLP/2023/Conference — EMNLP 2023 Main_

### Official Review · Reviewer_VXn3 · 2023-08-04

**Soundness:** 4

**Excitement:**

4: Strong: This paper deepens the understanding of some phenomenon or lowers the barriers to an existing research direction.

**Missing References:**

This study seems to miss many Technology Assisted Review (TAR) references. For example, you may find the following to be helpful:

Di Nunzio, Giorgio Maria, and Evangelos Kanoulas. "Special Issue on Technology Assisted Review Systems." Intelligent Systems with Applications (2023): 200260.

Kanoulas, Evangelos, Dan Li, Leif Azzopardi, and Rene Spijker. "CLEF 2019 technology assisted reviews in empirical medicine overview." In CEUR workshop proceedings, vol. 2380, p. 250. 2019.

Cormack, G.V. and Grossman, M.R., 2016, July. Engineering quality and reliability in technology-assisted review. In Proceedings of the 39th International ACM SIGIR conference on Research and Development in Information Retrieval (pp. 75-84).

**Paper Topic And Main Contributions:**

This study guided the future of technology assisted reviews in health and medicine by interviewing international systematic review experts to characterize the perceived utility and risks of large language models (LLMs) in the specific context of medical evidence reviews

**Questions For The Authors:**

Question A: Would you be able to release the interview data their codification, or other additional resources related to the interview data and its analysis without compromising the ethics protocol and participant consenting?

**Reasons To Accept:**

* Likely to be an influential study going forward because the paper addressed the perceived utility and risks of the breakthrough innovation of LLMs to assist authoring medical evidence reviews

* Excellent and insightful methodology: experts were interviewed

* The paper is excellently written, and this writing demonstrated authors' methodological excellence in the review topic as well (e.g., LLMs, systematic reviews, and qualitative analysis)

**Reasons To Reject:**

* Interview transcripts, their codification, or other additional resources related to the interview data and its analysis do not seem to be released together with the paper. They would have highlighted the methodological rigor, improved result transparency, and encouraged future work.

* The extent of new insights related to the perceived utility of technology assisted reviewing (TAR) is quite limited compared to the prior work, noting that TAR protocols, guidelines, and shared tasks have been published and organised for at least 5-10 years

**Reproducibility:**

N/A: Doesn't apply, since the paper does not include empirical results.

**Reviewer Confidence:**

4: Quite sure. I tried to check the important points carefully. It's unlikely, though conceivable, that I missed something that should affect my ratings.

**Typos Grammar Style And Presentation Improvements:**

* Introduce all acronyms (e.g., LLMs in the abstract)
* In-text citation lists should be organised from the oldest to the latest (and alphabetically within a given year) (e.g., "(Murad et al., 2016; Cook et al., 1997; Mulrow, 1987)" should be "(Mulrow, 1987; Cook et al., 1997; Murad et al., 2016)".

---

> ### Author Rebuttal · Authors · 2023-08-25
>
> We thank the reviewer for their constructive, thoughtful review, and address the main concerns below.
>
> **Making Interview Transcripts and Their Codification Available**
>
> We agree that releasing the interview transcripts and their codification would be ideal and would provide improved transparency for our analysis. However, as anticipated by the reviewer, this would compromise the protocol we presented to the IRB prior to engaging in this exercise. We designed our protocol this way to allow participants to talk freely without fear of views being attributed to their organizations (since many held senior positions in academia, guidelines, and evidence synthesis). That said, we will talk with our IRB to see if we can release redacted transcripts and their codification (after seeking permission from individual participants); but we cannot promise that these will be released at this time (as it will depend on their determination).
>
> **Limited References to Technology Assisted Review (TAR)**
>
> In the paper, we focused on presenting our work as the first qualitative assessment of LLMs for the task of drafting medical systematic reviews with experts, which is complementary to prior work on technology assisted reviews. However, we are grateful for the additional pointers and will add a discussion of these prior efforts to further contextualize our contribution.
>
> **Presentation Improvements**
>
> Thank you for suggesting concrete ways for us to improve the presentation of our work. We will make sure to spell out all acronyms when used for the first time in the paper and to organize the in-text citation lists in chronological order.

---

### Official Review · Reviewer_UT4H · 2023-08-15

**Soundness:** 3

**Excitement:**

4: Strong: This paper deepens the understanding of some phenomenon or lowers the barriers to an existing research direction.

**Missing References:**

[These are contemporaneous work, hence not required to be cited]

1. Qureshi, R., Shaughnessy, D., Gill, K.A.R. et al. Are ChatGPT and large language models “the answer” to bringing us closer to systematic review automation?. Syst Rev 12, 72 (2023). https://doi.org/10.1186/s13643-023-02243-z

2. Sallam, M. ChatGPT Utility in Healthcare Education, Research, and Practice: Systematic Review on the Promising Perspectives and Valid Concerns. Healthcare 2023, 11, 887. https://doi.org/10.3390/healthcare11060887

**Paper Topic And Main Contributions:**

The paper explores the potential of using LLMs to generate medical evidence reviews as well as highlights the risk of LLMs generating inaccurate or misleading content. To address this, the authors interviewed systematic review experts, who shared that LLMs could aid in drafting summaries and templates but expressed concerns about inaccuracies and reduced accountability. The paper contributes by outlining evaluation criteria for biomedical LLMs, aligning with expert perspectives.

**Reasons To Accept:**

1. The paper is novel as this study is the first attempt to characterize "expert" views on LLMs for the drafting of medical systematic reviews.
2. This study has the potential of enhancing the reliability of LLM-generated evidence summaries.
3. The article has a clear narrative and makes it clear to the reader what message the article is trying to convey.

**Reasons To Reject:**

1. The authors tested three LLM models with various prompts, but details on selecting prompts for specific models and the process of prompting experimentation are lacking.

2. The authors only shared a small number of samples (potentially just three) during the interview with participants, and one of these samples was chosen based on the participant's background. This limited number of samples and the small participant size could impact the representativeness of the findings.

**Reproducibility:**

N/A: Doesn't apply, since the paper does not include empirical results.

**Reviewer Confidence:**

4: Quite sure. I tried to check the important points carefully. It's unlikely, though conceivable, that I missed something that should affect my ratings.

---

> ### Author Rebuttal · Authors · 2023-08-25
>
> We thank the reviewer for the constructive, thoughtful reviews and for acknowledging the novelty and contribution of our work. Below we focus on addressing the main concerns about our work.
>
> **Limited Prompting Experimentation**
>
> We will provide more details on the prompt selection for each model in Appendix C.1. In regards to the lack of prompting experimentation, we are aware that our very basic prompting strategy might have failed to leverage the full potential of LLMs for generating medical systematic reviews. In this study, we did not focus on doing a head-to-head comparison of different prompting strategies but used basic prompts to generate outputs to demonstrate the full capabilities of the benefits and harms these LLMs can do.
>
> **Small Number of Samples and Participant Size**
>
> Qualitative methods do not aim to provide statistical generalisability; and instead rely on in-depth, unstructured data from a smaller number of diverse participants, aiming to uncover a full range of themes. Our sample size is typical for the research method. We acknowledge, as with any qualitative research, that additional interviews may have generated new ideas or themes for analysis. Similarly, our goal in providing LLM examples was to familiarize our participants (most of whom did not have deep LLM experience) with typical outputs, limitations, and opportunities of these models. We do not aim to provide a definitive accuracy evaluation of these specific models; instead, we aim to learn about general issues in their application. We highlight that our conclusions focus on generally applicable issues, and complement other quantitative work on individual models/approaches.
>
> **Missing References**
>
> Thank you for bringing these contemporaneous related works to our attention. We will add them to the Related Work section of our paper.

---

### Official Review · Reviewer_w8DF · 2023-08-15

**Soundness:** 3

**Excitement:**

3: Ambivalent: It has merits (e.g., it reports state-of-the-art results, the idea is nice), but there are key weaknesses (e.g., it describes incremental work), and it can significantly benefit from another round of revision. However, I won't object to accepting it if my co-reviewers champion it.

**Paper Topic And Main Contributions:**

This paper offers a comprehensive and in-depth qualitative analysis that delves into the systematic review generation capabilities of three distinct Language Model Models (LLMs). The approach adopted by the authors involves conducting interviews with domain experts, aiming to uncover the potential utilities and risks associated with LLMs in the context of producing medical system reviews. Additionally, expert opinions have been gathered to explore strategies for enhancing the reliability of LLMs for medical systematic reviews.

**Reasons To Accept:**

1. The authors have engaged in insightful interviews with domain experts to meticulously evaluate the advantages and potential risks linked to the utilization of LLMs in the domain of medical systematic reviews. These interviews have yielded invaluable perspectives on a spectrum of aspects, ranging from the potential applications of LLMs in medical systematic reviews to concerns and potential harms tied to their unwarranted application. Additionally, the interviews offer constructive insights into strategies for enhancing the reliability of LLMs, thereby contributing to bridging the gap associated with their implementation. In each of these cases, authors have backed up their claims with examples from domain experts' comments.

2. The authors have identified 11 general concepts characterizing model-generated summaries.

3. The paper includes all the relevant details regarding recruitment of domain experts and interview process which is a great contribution to the research domain as it can be used as reference for further related studies.

**Reasons To Reject:**

1. In this study, three distinct LLMs named Galactica, BioMedLM, and ChatGPT have been selected by the authors. The differences between these LLMs are outlined in the related work section of the paper. Despite their notable distinctions in training data and size, the evaluation of all these LLMs follows a uniform approach. A more effective approach would involve assessing the outputs of each LLM separately, as this could provide valuable insights into their relative performance in generating medical systematic reviews. It is plausible that certain LLMs might exhibit higher risk factors, while others could excel in generating coherent systematic reviews. In essence, this study would benefit significantly from a comprehensive comparative analysis between the LLMs, allowing for a more nuanced understanding of their respective capabilities, limitations, and potential benefits.

2. The number of samples presented to each domain expert appears to be relative inadequate to draw definitive conclusions about the abilities and constraints of LLMs in generating systematic reviews. Additionally, during the expert interviews, the inclusion of human-written systematic reviews, without indicating their human origin, could offer valuable insights. This approach would allow observation of how domain experts react to these reviews, shedding light on the deficiencies of LLM-generated systematic reviews and thereby allowing a more comprehensive understanding of the lacking of the LLM generated review.

3. The prompting technique used in this study is very basic and fail to leverage the full potentials of LLMs. Carefully curated prompts can gain better results in generating better systematic reviews.

**Reproducibility:**

N/A: Doesn't apply, since the paper does not include empirical results.

**Reviewer Confidence:**

3: Pretty sure, but there's a chance I missed something. Although I have a good feel for this area in general, I did not carefully check the paper's details, e.g., the math, experimental design, or novelty.

---

> ### Author Rebuttal · Authors · 2023-08-25
>
> We thank the reviewer for the constructive, thoughtful review and for acknowledging how our findings provide valuable perspectives on the benefits, harms, and risks of using large language models (LLMs) for medical systematic reviews. Below we address the main concerns raised about our work.
>
> **Comparative Analysis of LLMs**
>
> We agree that conducting a comprehensive quantitative comparative analysis of LLMs in this application space would be a valuable exercise. We also agree that the performances of different LLMs will likely differ due to various factors (training data, model size, architecture, etc.). However, this qualitative work is not intended to provide such comparison; rather, here we are looking at LLMs as a general new technology with associated risks and benefits which we believe will be similar for all current state-of-the-art models. To this end, we used 3 representative LLMs to generate examples to catalyze discussion in our interviews with experts, as many have not used or seen outputs from LLMs for the medical systematic review generation task. We highlight that our main conclusions call attention to issues (and opportunities) across all current generation LLMs.
>
> As we do not aim to provide a direct head-to-head comparison of models, we have explicitly noted this in the Limitations section: “Since this study was not a performance evaluation of LLMs but an exploratory study to understand the views of domain experts, further empirical studies are needed to actually evaluate the effectiveness of LLMs for medical systematic reviews.” In sum: comparing different LLMs for this task would be a useful undertaking, but it is not the focus of this qualitative work which aims to assess expert perceived potential utility and risks of LLMs—as a general technology—for systematic reviews.
>
> **Small Number of Samples**
>
> We used a qualitative method, aiming to discover a diversity of themes (rather than aiming for statistical representativeness). Qualitative research allows for richer, more detailed analysis of unstructured data from a smaller number of participants. Our sample size is typical for this methodology.
> Domain expert time is scarce, and in-depth interviews about references and automatically generated evidence summaries are inherently time-intensive and therefore limit the possible scale of the analysis. We acknowledge that conducting additional interviews might have generated additional themes.
>
> **Very Basic Prompting Technique**
>
> We agree that our basic prompting strategy might have failed to leverage the full potential of LLMs for generating medical systematic reviews. However, our focus here was not on performing a comparative analysis of prompting strategies; our aim was merely to provide illustrative examples to familiarize the experts with typical outputs, shortcomings, and strengths of current generation LLMs. Our conclusions highlight broad thematic issues found empirically in representative LLMs. We anticipate that improved prompts will not greatly affect these broad thematic areas.

---

### Meta-Review · Area_Chair_eNXV · 2023-09-14

**Recommendation:** 4

**Metareview:**

This paper provides a qualitative study of how experts in systematic reviews perceive the potential uses and drawbacks of using LLMs to conduct systematic literature reviews in medical research. All reviewers agree that this is a well-written and well-motivated paper tackling an important issue in a robust way. The reviewers agree that the insights derived from the qualitative data are novel and insightful and likely to be impactful for future medical researchers and practitioners. I concur with the reviewers, particularly given the prevalence of medical domain experts (and domain experts from law, education, policy, etc) using LLMs to support research, it is critical to understand how such practitioners perceive the capabilities and limitations of LLMs for such tasks.

Reviewers wanted to see a comparative analysis of the performance of different LLMs, and raised concerns about the number of samples, both of which the authors addressed in their rebuttal. The authors should add additional context about the qualitative data collection and analysis methods used to further justify the approach taken.

Reviewers raised concerns about the prompting approach taken, which the authors satisfactorily responded to by justifying their approach - that experts in systematic reviews who are turning to LLMs to support their work may not necessarily have expertise in prompt engineering, and thus a qualitative approach that maintains fidelity to the practitioners’ typical work practices is a reasonable approach to take (Salvagno et al., 2023), rather than one that may adopt state of the art practices in prompt engineering, which medical practitioners may not themselves adopt. To address this, I recommend that the authors better contextualize that the qualitative approach taken here is designed to have ecological validity with respect to medical practitioners’ use of LLMs for systematic reviews and the tradeoffs of such use within the social context of medical research practice, rather than making claims about the technical capability of LLMs divorced from such social context (where, for instance, using SotA prompting techniques and conducting comparative analyses of various model capabilities would be more appropriate).

Reviewers raised concerns about the sample size, however, for qualitative interview studies such as this, such sample sizes are well within the bounds of acceptable sample sizes. Prior research suggests that sample size is not a sufficient grounds on which to evaluate rigor of qualitative research and should be contextualized with respect to other approaches to establishing validity of the results (Caine, 2014).


https://ccforum.biomedcentral.com/articles/10.1186/s13054-023-04380-2

---

### Decision · Program_Chairs · 2023-10-07

**Decision:**

Accept-Main

**Comment:**

This paper provides a qualitative study of how experts in systematic reviews perceive the potential uses and drawbacks of using LLMs to conduct systematic literature reviews in medical research. All reviewers agree that this is a well-written and well-motivated paper tackling an important issue in a robust way. The reviewers agree that the insights derived from the qualitative data are novel and insightful and likely to be impactful for future medical researchers and practitioners. I concur with the reviewers, particularly given the prevalence of medical domain experts (and domain experts from law, education, policy, etc) using LLMs to support research, it is critical to understand how such practitioners perceive the capabilities and limitations of LLMs for such tasks.

Reviewers wanted to see a comparative analysis of the performance of different LLMs, and raised concerns about the number of samples, both of which the authors addressed in their rebuttal. The authors should add additional context about the qualitative data collection and analysis methods used to further justify the approach taken.

Reviewers raised concerns about the prompting approach taken, which the authors satisfactorily responded to by justifying their approach - that experts in systematic reviews who are turning to LLMs to support their work may not necessarily have expertise in prompt engineering, and thus a qualitative approach that maintains fidelity to the practitioners’ typical work practices is a reasonable approach to take (Salvagno et al., 2023), rather than one that may adopt state of the art practices in prompt engineering, which medical practitioners may not themselves adopt. To address this, I recommend that the authors better contextualize that the qualitative approach taken here is designed to have ecological validity with respect to medical practitioners’ use of LLMs for systematic reviews and the tradeoffs of such use within the social context of medical research practice, rather than making claims about the technical capability of LLMs divorced from such social context (where, for instance, using SotA prompting techniques and conducting comparative analyses of various model capabilities would be more appropriate).

Reviewers raised concerns about the sample size, however, for qualitative interview studies such as this, such sample sizes are well within the bounds of acceptable sample sizes. Prior research suggests that sample size is not a sufficient grounds on which to evaluate rigor of qualitative research and should be contextualized with respect to other approaches to establishing validity of the results (Caine, 2014).


https://ccforum.biomedcentral.com/articles/10.1186/s13054-023-04380-2